# Modelled ocean changes at the Plio-Pleistocene transition driven by Antarctic ice advance

Daniel J. Hill[1], Kevin P. Bolton[2] & Alan M. Haywood[1]

The Earth underwent a major transition from the warm climates of the Pliocene to the Pleistocene ice ages between 3.2 and 2.6 million years ago. The intensification of Northern Hemisphere Glaciation is the most obvious result of the Plio-Pleistocene transition. However, recent data show that the ocean also underwent a significant change, with the convergence of deep water mass properties in the North Pacific and North Atlantic Ocean. Here we show that the lack of coastal ice in the Pacific sector of Antarctica leads to major reductions in Pacific Ocean overturning and the loss of the modern North Pacific Deep Water (NPDW) mass in climate models of the warmest periods of the Pliocene. These results potentially explain the convergence of global deep water mass properties at the Plio-Pleistocene transition, as Circumpolar Deep Water (CDW) became the common source.

[1] School of Earth and Environment, University of Leeds, Woodhouse Lane, Leeds LS2 9JT, UK. [2] Environment and Sustainability Institute, College of Engineering, Mathematics and Physical Sciences, University of Exeter, Penryn Campus, Cornwall TR10 9FE, UK. Correspondence and requests for materials should be addressed to D.J.H. (email: eardjh@leeds.ac.uk).

Today, the world's major deep water masses, found in the depths of the Pacific and Atlantic Oceans, are relatively homogenous. This is primarily because they are connected through the Circumpolar Deep Water (CDW) of the Southern Ocean. The CDW allows deep water masses produced in different parts of the globe, primarily in the Weddell and Ross Seas of coastal Antarctica and different regions of the North Atlantic, to mix together before being exported to both the deep Atlantic and Pacific Oceans[1]. A recent comparison of long benthic records of sea water oxygen isotopes from the North Pacific and North Atlantic shows that the records converge at the same time as the intensification of Northern Hemisphere glaciation. This also corresponds with the convergence of Atlantic and Pacific records of the oxygen isotopes of calcite and bottom water temperatures. Although Atlantic sea water oxygen isotope values stay within a similar range through the Pliocene and Pleistocene, the record from the North Pacific moves from isotopically lighter values to approach the Atlantic values[2]. Bottom water temperature variations in the Pacific and Atlantic also synchronize over this transition, suggesting that the North Pacific no longer responds to local forcings, but to a global signal, after the transition. As the Southern Ocean is the primary connection between the Pacific and Atlantic, and large changes in global ice cover are known to have occurred during the Plio-Pleistocene transition, Antarctic glaciation could be suggested as a driver of this change. The North Pacific underwent massive changes during the Plio-Pleistocene transition, with the instigation of permanent stratification, significantly decreased biological productivity, changes in the biological pump efficiency and probably a lowering in the carbon flux from ocean to atmosphere[3].

The interval of enhanced warmth preceding the Plio-Pleistocene Transition, the mid-Piacenzian Warm Period (mPWP) of 3.264–3.025 Ma[4], has been well studied using climate models, culminating in the Pliocene Model Intercomparison Project (PlioMIP)[5]. The PlioMIP experimental boundary conditions impose a number of changes in the Pliocene simulations, including atmospheric carbon dioxide concentrations, orography, vegetation, sea level and ice sheet changes[6]. Here, to examine ocean circulation and try to simulate the changes seen in the North Pacific, we use this experimental design[6] within the Hadley Centre coupled ocean–atmosphere climate model (HadCM3; see Method). As one of the most likely to affect global ocean circulation and the change implicated in records of North Pacific sea water isotope variations[2], we perform a series of sensitivity simulations examining the effect of the changes in the Antarctic Ice Sheet on circulation in the Pacific Ocean. In addition to the PlioMIP standard simulation, two further HadCM3 simulations are performed. In the first, a modern East Antarctic Ice Sheet (EAIS) is imposed, while maintaining an ice-free West Antarctic Ice Sheet (WAIS; and the other PlioMIP boundary conditions). A second simulation imposes a modern ice sheet configuration over both East and West Antarctica, in an otherwise standard PlioMIP simulation (Fig. 1). These simulations show that Antarctic ice advance after the mPWP, in contrast to previous suggestions[2,7], causes reductions in Pacific Ocean circulation.

## Results

### Pacific Ocean meridional overturning circulation.
The present day deep Pacific Ocean is filled with Antarctic sourced water, known as Pacific Deep Water (PDW). Much of this water returns to the south from the tropics at intermediate depths to the CDW, but significant volumes of deep water penetrate northward, forming the North PDW (NPDW). There is no significant

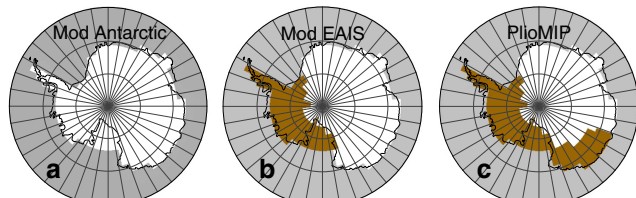

**Figure 1 | Different Antarctic Ice Sheet boundary conditions used within the HadCM3 climate model.** White areas are ice covered, grey areas are ocean and brown areas are ice free. Modern experiments use a continent wide ice sheet (**a**), a reconstruction with fully retreated West Antarctica and a modern East Antarctica (**b**), whereas significant retreat of both West Antarctica and East Antarctica is specified in the PlioMIP boundary conditions[60] (**c**).

deep water formation in the North Pacific, so the NPDW fills even the northernmost deep Pacific. The HadCM3 Pre-industrial simulation reproduces this large-scale Pacific Ocean meridional circulation (Fig. 2a).

In the PlioMIP standard simulation, large changes occur in Pacific Ocean circulation. PDW circulation is slightly reduced, but very little NPDW is produced and does not penetrate nearly as far (~20° less far north) into the North Pacific (Fig. 2b). Reintroducing the present day Antarctic Ice Sheet, to an otherwise Pliocene simulation, returns Pacific Ocean circulation to close to its present day configuration, although PDW is still marginally weaker (Fig. 2c). With just the WAIS retreated and a modern EAIS, the Pacific circulation is increased from the PlioMIP standard, although still with significant reductions from pre-industrial (Fig. 2d).

In these simulations, the presence (or absence) of NPDW is related to the presence (or absence) of a coastal ice sheet in the Pacific sector of the Antarctic (both the WAIS[8] and the EAIS, incorporating Northern Victoria Land[9], Wilkes Subglacial Basin[10] and Aurora Subglacial Basin[11]). This change is corroborated by the PlioMIP model ensemble, which shows that, given the same boundary conditions, other climate models also show a weakening, or indeed a reversal of, NPDW circulation (Table 1). The magnitude of the change varies between different members of the PlioMIP ensemble, but the sign of the change is consistent.

### Pliocene Southern Ocean changes.
If changes in Antarctica are being seen in the deep North Pacific, which is largely isolated from the atmosphere, then it seems clear that there must be changes in the Southern Ocean propagating this effect. Southern Ocean circulation is dominated by CDW and the return flow of deep waters from the Pacific and Atlantic Oceans. The other important feature of the circulation is the export of Antarctic Bottom Water (AABW), which forms close to the Antarctic continent and flows into the deepest parts of the Pacific and Atlantic Oceans. This occurs at a select number of latitudes and locations around coastal Antarctica, particularly in the Ross and Weddell Seas, so it is the overturning associated with its surface formation and northward export in the deep ocean that form a coherent negative signal on zonally averaged meridional overturning circulation (MOC) plots (Fig. 3a). In the Pliocene standard simulation, all of the components of Southern Ocean circulation have a reduced strength (Fig. 3b). This shows that in the Pliocene there was reduced AABW formation and a lower volume of CDW waters flowing back to the Southern Ocean. Although the Atlantic sector contributes to these circulation changes, the differences are dominated by changes in the

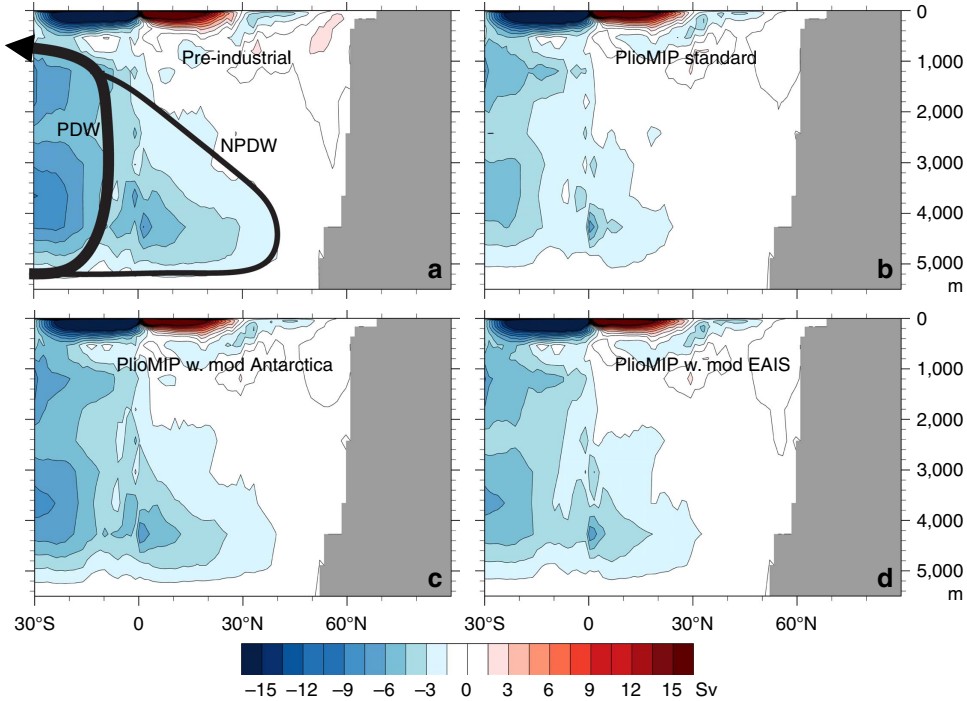

**Figure 2 | MOC in the North Pacific from the HadCM3 simulations.** Pre-industrial MOC (**a**) shows schematically the major Pacific deep water masses, the PDW and the NPDW. The PlioMIP standard simulation (**b**) shows that the Pliocene NPDW is much weaker and does not penetrate into the North Pacific above 30°N. Introducing a modern Antarctic Ice Sheet (**c**) largely reintroduces modern Pacific MOC, while a modern EAIS (with an ice free WAIS; **d**) only marginally increases Pacific MOC. Positive values represent clockwise overturning, whilst negative values represent anti-clockwise circulation.

### Table 1 | Analysis of PlioMIP simulations.

| Simulation | Pre-industrial (Sv) | Pliocene (Sv) | Abs. change (Sv) | Change in Pacific sector wind strength (%) |
|---|---|---|---|---|
| CCSM4 | −1.93 | −0.67 | −1.26 | +3.7 |
| COSMOS | −1.81 | 1.61 | −3.42 | +62.8 |
| FGOALS-g2 | −2.05 | 1.12 | −3.17 | +36.0 |
| HadCM3 | −2.59 | −1.06 | −1.56 | +26.4 |
| IPSLCM5A | −12.6 | −12.0 | −0.6 | +94.4 |
| MIROC4m | −3.74 | −3.25 | −0.49 | +29.2 |
| MRI-CGCM 2.3 | −12.6 | −12.0 | −0.6 | +45.1 |
| NorESM-L | −2.51 | −1.36 | −1.15 | +76.5 |

CCSM4, Community Climate System Model 4: National Center for Atmospheric Research, USA; COSMOS, Alfred Wegener Institute, Bremerhaven, Germany; FGOALS-g2, Flexible Global Ocean–Atmosphere–Land System model grid-point version; g2, State Key Laboratory of Numerical Modeling for Atmospheric Sciences and Geophysical Fluid Dynamics (LASG), Institute of Atmospheric Physics (IAP), Chinese Academy of Sciences (CAS), China; HadCM3, Hadley Centre Coupled Model 3, UK Met Office Unified Model, UK; IPSLCM5A, Laboratoire des Sciences du Climat et de l'Environnement (LSCE), France; MIROC4m, Center for Climate System Research, University of Tokyo, Japan; MOC, meridional overturning circulation; MRI-CGCM 2.3, Meteorological Research Institute and University of Tsukuba, Japan; NorESM-L, Norwegian Earth System Model: Bjerknes Centre for Climate Research, Bergen, Norway; PlioMIP, Pliocene Model Intercomparison Project.
Meridional overturning circulation (Sv = $10^6$ m³ s⁻¹) in the deepest part of the North (>30°N) Pacific for PlioMIP simulations. Absolute changes show a robust increase in anti-clockwise (negative) circulation in the pre-industrial simulations compared to the Pliocene. The final column shows the associated increase in the mean wind strength in the Pacific sector of the Southern Ocean in the pre-industrial simulations compared to the Pliocene.

Pacific sector of the Southern Ocean. Reintroducing the Antarctic Ice Sheet in the PlioMIP simulation largely returns the Southern Ocean circulation to its Pre-industrial state, especially the AABW formation and export (Fig. 3c). With just a modern EAIS, AABW formation remains close to pre-industrial levels (whereas there are greater changes in the CDW and the deep Southern Ocean), although still significantly less than the PlioMIP simulation (Fig. 3d). These changes are in contrast to the mechanisms for Antarctica driving North Pacific changes suggested by Woodard et al.[2], as they invoked greater deep water formation[7] to explain the convergence of North Pacific and North Atlantic records. Although there are changes in shallower depths associated with the other model boundary condition changes[6] in the Pliocene (Fig. 3c), these sensitivity simulations show that Southern Ocean circulation is largely responding to the changes in the Antarctic Ice Sheet.

Antarctic sea ice plays an important role in Southern Hemisphere deep water formation, through brine rejection and heat losses[12]. In the warm Pliocene, there was significantly less sea ice around Antarctica[13–15] and our simulations show that this is due to both the loss of the ice sheet around the Antarctic coast and the warming due to other Earth system changes during the mPWP (Fig. 4). Although there is a general correlation between colder temperatures, increased ice sheet extent and greater sea ice, geographically the response is more complicated.

**Mechanism for Antarctic ice to alter Pacific circulation.** The advance of ice to the Antarctic coast in the Pacific sector (Fig. 1) increases the wind strength (Fig. 5a) and the wind stress on the ocean (Fig. 5b) that drives the Antarctic Circumpolar Current

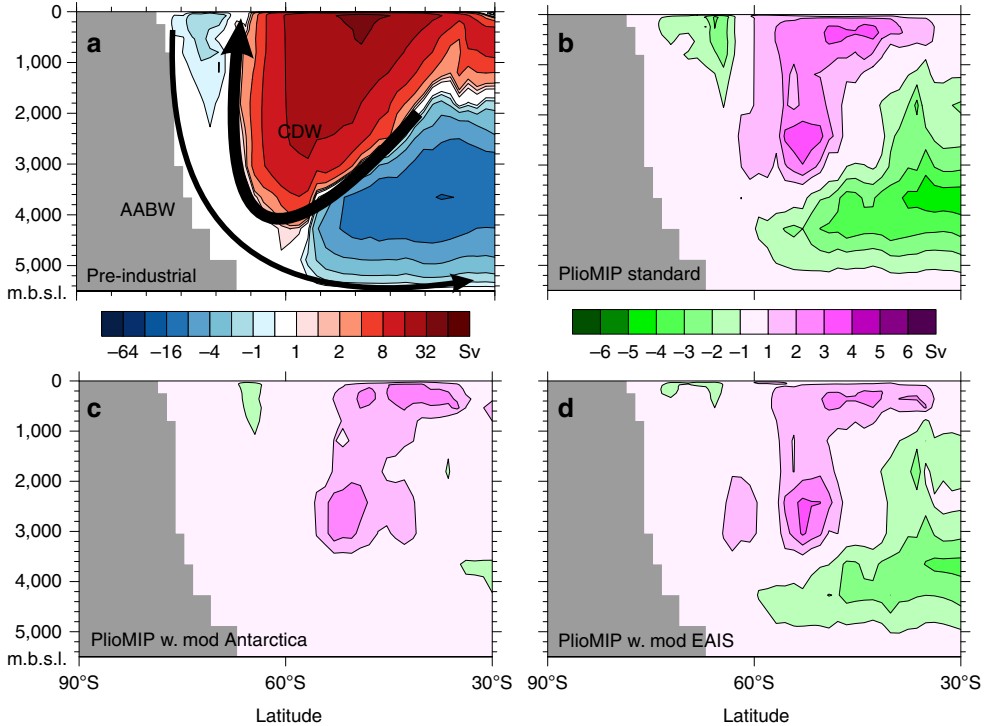

**Figure 3 | MOC in the Southern Ocean.** Pre-industrial MOC (**a**) shows schematically the major Southern Ocean deep water masses, the CDW and the AABW, integrated over all longitudes, from 90°S to 30°S over the full depth of the ocean, in metres below sea level (m.b.s.l.). The Pliocene anomalies (**b–d**) show the changes that would have occurred at the Plio-Pleistocene transition (that is, pre-industrial minus Pliocene). The PlioMIP standard simulation (**b**) shows that CDW is weaker, and that AABW formation and export is significantly reduced. Introducing a modern Antarctic Ice Sheet (**c**) returns the AABW to its pre-industrial state and greatly reduces differences in the CDW. With a modern EAIS (**d**) there is some further response in the CDW and AABW export, but very little change in AABW production. Positive values represent clockwise overturning (or possibly less anticlockwise overturning in anomaly plots), whereas negative values represent anti-clockwise circulation (or possibly less clockwise circulation in anomaly plots). In these anomaly plots, as the anomalies generally have the same sign as the absolute, show the spinning up of circulation at the Plio-Pleistocene transition.

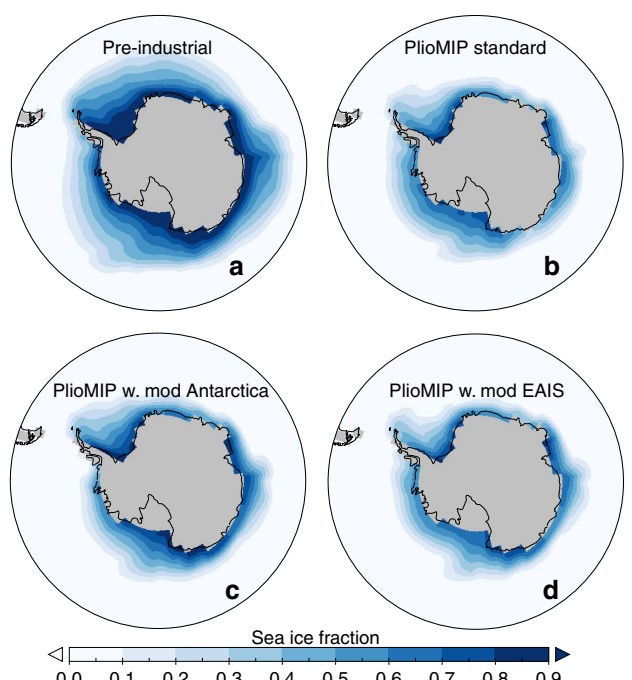

**Figure 4 | Annual mean Antarctic sea ice fraction in each of the HadCM3 simulations.** Pre-industrial (**a**), PlioMIP standard (**b**), PlioMIP with modern Antarctic Ice Sheet (**c**) and PlioMIP with modern EAIS (**d**), showing a general trend of decreasing sea ice as Pliocene ice extents are reached, but with significant spatial heterogeneity.

in this region of the Southern Ocean and the coastal Antarctic counter-current in the Pacific and Indian Ocean sectors. This atmospheric forcing is an important driver of upwelling and bottom water production in the Southern Ocean and a key component of global ocean circulation[16]. The geographic extent of the transfer of this surface forcing to the deeper ocean is best seen through the increases in the mixed layer depth (Fig. 5c), with the increased strength of overturning leading to greater export of CDW and AABW into the deep Pacific Ocean (Fig. 3) and ultimately to the production of NPDW (Fig. 2). As the NPDW has its origins in the CDW, it is the initiation of this Pacific water mass that caused the homogenization of the North Atlantic and NPDW properties[2]. In addition to showing reduction in MOC in the North Pacific, the PlioMIP ensemble also shows a consistent increase of winds in the Pacific sector of the Southern Ocean (Table 1), suggesting that similar changes are occurring across the suite of models in the ensemble.

**Comparisons with Plio-Pleistocene records from the Pacific.** Climate models do not routinely incorporate water isotopes or tracers into their simulations and therefore direct comparisons with records of ocean circulation are difficult. Although subject to a number of other dependencies and ocean circulation changes[17], differences in Pacific Ocean carbon isotope records between the Pliocene and Pleistocene interglacials do schematically reflect what would be expected from the simulations presented here. In equatorial Pacific bottom waters, there is no change in carbon isotopes. Higher in the water column, in the path of modern day southward flowing NPDW return waters, significant

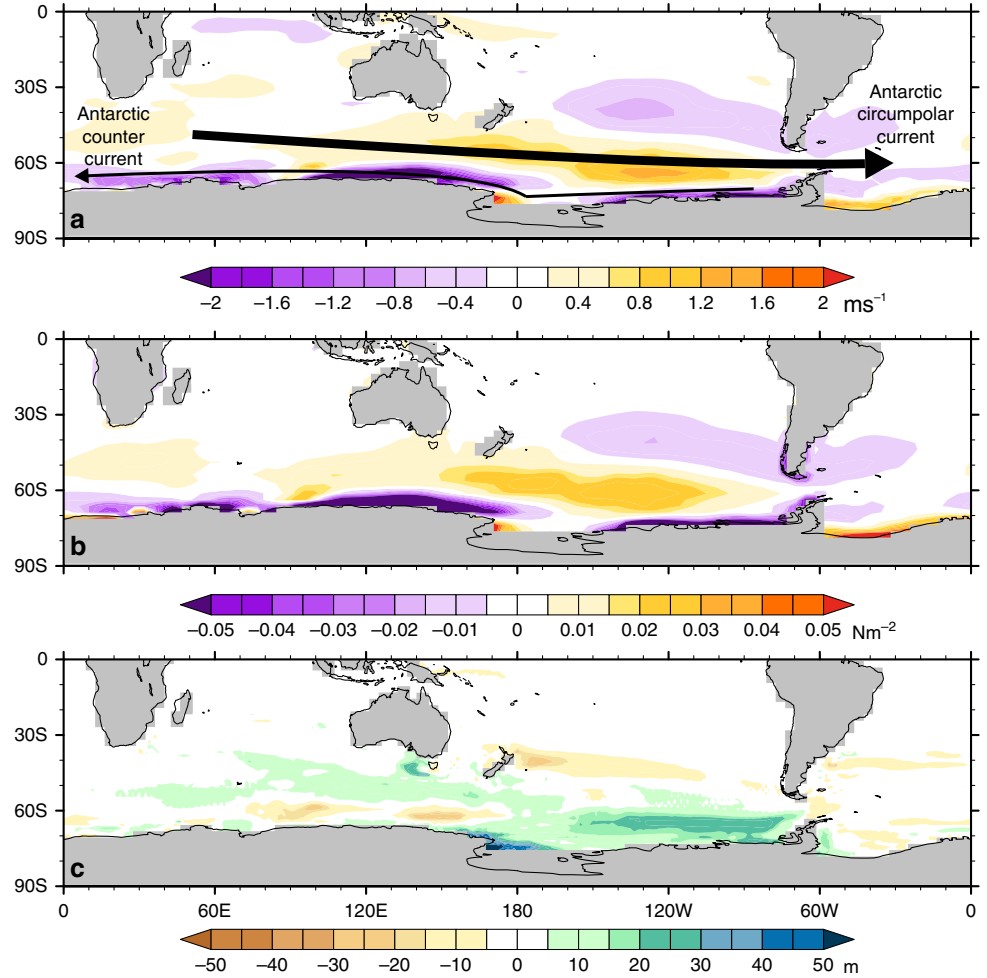

**Figure 5 | Modelled mechanism by which Antarctic ice advance changes ocean circulation.** All fields shown are the difference between PlioMIP with modern Antarctic Ice Sheet and PlioMIP standard simulations. The arrows depict the general location of the Antarctic Circumpolar Current (upper, thick arrow) and the Antarctic counter current (lower, narrow arrow). The advance of Antarctic ice in the Pacific sector (Fig. 1) causes the westerly winds to increase (positive values) in the Pacific sector of Southern Ocean (and not the Atlantic sector). Close to the Antarctic coast, the easterly winds, which help drive the Antarctic counter-current, also increase (negative values) (**a**). These wind strength increases translate into increased wind stress on the ocean (**b**), which increases the overturning in the Pacific sector, shown through the modelled mixed layer depth (**c**), which then invigorates Pacific MOC and the export of NPDW (Fig. 2).

**Table 2 | Modelled deep ocean warming in the Pacific and Atlantic Ocean.**

| Site | Pre-industrial (°C) | PlioMIP standard (°C) | PlioMIP w. mod Antarctica (°C) | PlioMIP w. mod EAIS (°C) |
|---|---|---|---|---|
| ODP 1208 (North Pacific) | 0.49 | 1.84 | 2.32 | 2.33 |
| DSDP 607 (North Atlantic) | 1.07 | 3.10 | 3.31 | 3.40 |

DSPD, deep ocean drilling project; EAIS, East Antarctic Ice Sheet; ODP, ocean drilling program; PlioMIP, Pliocene Model Intercomparison Project.
Simulated ocean temperatures at the sites used in the Woodard et al.[2] study to test Pacific and Atlantic gradients during the Plio-Pleistocene transition. Modelled in-situ temperatures are taken from the nearest grid box at the HadCM3 ocean layer at 3,347 m below sea level. DSDP Site 607 (41N, 32W) in the North Atlantic is at an ocean depth 3,427 m and ODP Site 1,208 at Shatsky Rise (36.13N, 158.2W) in the North Pacific is at a depth of 3,350 m.

differences in carbon isotopes were found (0.19–0.24‰)[18]. These seem to accord with the simulations showing little change in equatorial deep water circulation, but a loss of NPDW.

Temperatures are not an ideal tracer of ocean circulation, as there are many processes that produce deep ocean temperature change over geological timescales[19]. Our model simulations do show a reduction of the temperature difference between the North Atlantic and North Pacific from the Pliocene to the pre-industrial (Table 2). However, the simulated difference in the

Pliocene is not as enhanced as seen in the records[2], perhaps suggesting that there could be local palaeogeographic changes not incorporated into the PlioMIP experimental design that affect the temperatures on Shatsky Rise[20–23]. As North Pacific circulation similar to modern is produced in the Pliocene simulations with more ice, this causes a warming of North Pacific waters, explaining the long-term increases in North Pacific temperatures during the initiation of Northern Hemisphere Glaciation[2].

In the Pacific sector of the Antarctic, a high resolution Plio-Pleistocene record was recovered by the ANDRILL program from sediments beneath the Ross Ice Shelf[8]. Over the transition the sediments show a reduction in sea surface temperatures and an increase in sea ice diatoms[9], matching the Antarctic sea ice changes simulated (Fig. 4). There are also increases in diatom species that favour polar open ocean or seasonal sea ice environments, evidencing the intensification of Antarctic coastal wind fields, the establishment of the Ross Sea polynya and probably its influence on southern sourced bottom water formation[13]. Although our simulations suggest much broader changes across the Pacific sector of the Southern Ocean, deep water formation in the Ross Sea would play a key role in the simulated changes (Fig. 5).

## Discussion

Antarctica underwent a climate transition in the latest Pliocene, becoming significantly cooler and exhibiting evidence for greater volumes of ice[13,24–26]. At this time, evidence from the ANDRILL core, underlying the modern Ross Ice Shelf, shows an increase in WAIS extent[8,13]. We show that the presence of a large WAIS and advances in the Pacific sector of East Antarctica[10,11,27], invigorated Pacific Ocean MOC, forming the NPDW extension of the global thermohaline circulation into the deep North Pacific basin. The export of CDW introduced a water mass that has been isolated from atmosphere for a long time to a large volume of the North Pacific. The impact of this previously unknown increase in carbon storage in the North Pacific on the overall ocean carbon storage potential must be balanced with the reduced overturning in the Southern Ocean and the possible increase in carbon release in the North Pacific[3]. However, the changes in the North Pacific could significantly increase the carbon stored and drive a geological rapid fall in atmospheric carbon dioxide concentrations. This change provides a plausible mechanism for Antarctica to drive the changes in the ocean at the Plio-Pleistocene transition and intensification of the Northern Hemisphere glaciation through reduced atmospheric $CO_2$ concentrations, which has already been shown to be the key factor in ice advance[28–30]. However, the overall impact of changes in Pacific Ocean circulation on the carbon cycle and especially atmospheric carbon dioxide levels needs to be quantified and modelling of these effects is beyond the scope of this modelling study and the capabilities of this modelling framework.

Temporary advances of the Antarctic Ice Sheet could have occurred during the glacial periods of the Pliocene[8,31,32] and reconstructions suggest that atmospheric $CO_2$ concentrations could have dropped close to or below pre-industrial levels[33–37]. The mechanisms presented here could explain at least part of the variability within the Pliocene. The WAIS was more stable after the Plio-Pleistocene Transition[8] and the evidence for total collapses remains sparse and poorly constrained[38–40], so the changes simulated here probably have a much smaller role to play in Pleistocene climate changes.

The climate transition at the Plio-Pleistocene boundary involves changes across a broad spectrum of Earth system components[41–43]. With changes in the Antarctic Ice Sheet, we would also expect changes in iceberg production[11], freshwater forcing[20] and Southern Hemisphere productivity[44], which have yet to be incorporated into models. Increased North Pacific carbon storage could lead to positive feedbacks from other carbon reservoirs[45,46], as they respond to the associated cooling, and therefore greater changes in atmospheric $CO_2$ than simply the carbon stored in the deep Pacific. The North Pacific does seem to play a significant role in glacial–interglacial $CO_2$ contrasts[47,48].

Evidence of the susceptibility of the WAIS and Pacific sector of East Antarctica to collapse under increasing temperatures and ongoing retreat in these areas continues to emerge[49–51]. The simulations presented here suggest that the oceanic changes seen at the Plio-Pleistocene Transition are reversible under Pliocene climate model boundary conditions, but it is not clear how future ice loss will change the deep ocean carbon reservoir.

## Methods

**HadCM3 version of the UKMO unified model.** The simulations run for this study use the UK Met Office coupled ocean–atmosphere General Circulation Model, HadCM3 (ref. 52). The atmosphere has a resolution of 3.75° longitude and 2.5° latitude, with 19 levels in the vertical. In the ocean, the model has 20 levels with a resolution of 1.25° by 1.25°. The radiation scheme follows the parameterization of Edwards and Slingo[53], the convection scheme of Gregory et al.[54] and the MOSES-1 land-surface scheme[55]. The ocean model uses the Gent and McWilliams[56] mixing scheme, coupled to a thermodynamic sea ice model with parameterized ice drift and sea ice leads[57]. Modern climate simulations have been shown to simulate sea surface temperatures in good agreement with observation, without requiring flux corrections[58] and have a good representation of ocean heat transport[52].

**Code availability.** Access to the Met Office Unified Model source code is available under license from the Met Office at http://www.metoffice.gov.uk/research/collaboration. Figures were prepared using Panoply and NCL free software packages, available from the National Aeronautics and Space Administration Goddard Institute for Space Studies (http://www.giss.nasa.gov/tools/panoply) and the National Center for Atmospheric Research (https://www.ncl.ucar.edu) respectively.

**Experimental design.** The simulations follow the alternative PlioMIP experiment 2 design of Haywood et al.[6], notwithstanding the changes in the Pliocene Antarctic Ice Sheet (Fig. 1). Atmospheric $CO_2$ concentrations are set to 405 p.p.m.v., consistent with the highest concentrations in the latest $CO_2$ reconstructions[33–37], vegetation cover is changed to the reconstruction of Salzmann et al.[42] and orography to the reconstruction of Sohl et al.[59].

The PlioMIP Standard simulation is a 500-year continuation of the original HadCM3 PRISM2 simulation, under altered PRISM3 boundary conditions[20], conforming to the PlioMIP experimental design simulation length. The pre-industrial simulation is similarly started from a long pre-industrial simulation and run for 500 years. Each of the sensitivity simulations run concurrently from the same starting point as the PlioMIP Standard, with the Antarctic boundary conditions changed. The fact that the PlioMIP w. Mod Antarctica simulation largely restores the pre-industrial Pacific meridional ocean circulation shows that the simulations are run for sufficient length to propagate the effects of the Antarctic boundary condition changes into Pacific overturning and that the changes seen in the Pliocene can be reversed.

The 8 PlioMIP Experiment 2 simulations were run with boundary conditions as close to those used in this study as possible, although some models used alternative versions[5]. The fact that the magnitude of Pacific MOC changes do not correlate with any of the key features of the simulations (for example, implemented boundary conditions, model resolution and parameterisations) suggests that the effect is a robust effect of the boundary condition changes.

**Data availability.** Climate model results are archived at the University of Leeds and the PlioMIP ensemble archive is hosted by the Bristol Research Initiative for the Dynamic Global Environment at the University of Bristol. Model data are available upon request to D.J.H. (eardjh@leeds.ac.uk).

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

## Acknowledgements

The PlioMIP analysis was undertaken by K.P.B. as part of a Natural Environment Research Council (NERC) Research Experience Placement (REP) hosted by the School of Earth and Environment at the University of Leeds, when an undergraduate at College of Engineering, Mathematics and Physical Sciences, University of Exeter, UK. A.M.H. acknowledges that his contribution to this research was funded by the European Research Council under the European Union's Seventh Framework Programme (FP7/2007–2013)/ERC grant agreement number 278636. Three anonymous reviewers are thanked for their comments, which produced an improved and more robust final manuscript.

## Author contributions

D.J.H. ran the HadCM3 simulations, analysed the results and wrote the manuscript, as well as designing and supervising the NERC REP project. K.P.B. analysed the PlioMIP

ensemble. A.M.H. leads the PlioMIP project, provided important feedback on the study and shaped the final manuscript.

## Additional information

**Competing financial interests:** The authors declare no competing financial interests.

**Publisher's note**: 

