## [Peer Review File · Nature Communications]

Reviewers' comments:

Reviewer #1 (Remarks to the Author):

In this paper Hill et al. present results from coupled ocean - atmosphere model simulations of the mid-Pliocene warm period (mPWP) with different configurations of the Antarctic ice sheet. They show that with mPWP boundary conditions an ensemble of models consistently simulate a reduction in the water inflow into the deep North Pacific compared to the present day. Using the HadCM3 model they attribute this change in ocean circulation to the reduced Antarctic ice volume and extent during the mPWP. The authors then describe the possible implications of this circulation change for the onset of Northern Hemisphere glaciation during the Pliocene-Pleistocene transition.

As acknowledged by the authors, the idea that an increase in the Antarctic ice volume could have affected the ocean overturning circulation has been already put forward by (Woodard et al., 2014) based on temperature and isotope reconstructions from two ODP sites in the North Atlantic and North Pacific. This paper adds a quantitative estimate of the possible contribution of changes in Antarctica to the overturning ocean circulation and describes possible mechanisms responsible for the circulation changes. The paper is definitely of interest for readers interested in the mechanisms that caused the Pliocene-Pleistocene transition. I therefore recommend publication if the authors can satisfactorily address some major comments outlined below.

Main comments

The results presented show a consistent weakening of the Pacific overturning circulation in idealized time-slab mPWP model simulations compared to present day. However, simulations of a single model are used to attribute the changes in Pacific circulation to the state of the Antarctic ice sheet. Could the authors elaborate on how representative the results are for the rest of the model ensemble? Without the need to run additional simulations, could the authors include some analysis of whether the pattern of changes in wind stress and mixed layer depth are a robust feature among the PlioMIP model ensemble or not?

The paper concentrates on the modeled differences in overturning circulation for an average 'interglacial' during the mPWP. However, both the mPWP and the subsequent Plio-Pleistocene transition are characterized by substantial orbital scale variability of both climate and ice sheets, particularly at the 40 kyr time scale corresponding to obliquity variations (Dolan et al., 2011; Lisiecki and Raymo, 2007; Prescott et al., 2014; Willeit et al., 2013). Thus the question arises whether the circulation changes described in the paper for the interglacials persisted also during glacial periods. Could the authors comment on that?

The last part of the abstract (lines 17-21) includes strong statements without any quantitative estimate of the effect of a circulation change on the mentioned processes: "This explains why the isotope records from the Pacific and Atlantic Oceans converge at the Plio-Pleistocene transition and suggests a novel explanation of the geologically rapid decrease in atmospheric carbon dioxide concentrations at the Plio-Pleistocene transition, the stratification of the North Pacific and the initiation of Northern Hemisphere glaciation." In particular I'm missing a quantitative estimate of the changes in deep North Atlantic and North Pacific temperatures induced by the circulation change. These could easily be compared to the reconstructed temperatures presented in (Woodard et al., 2014) and could give some support to the claim made by the authors that it was this change in overturning circulation that was responsible for the convergence of bottom water temperatures and (possibly) the isotopic records of the North Atlantic and Pacific.

The authors also argue that the increase in deep water inflow into the North Pacific could have increased the carbon stored in the ocean thereby causing the atmospheric CO₂ to drop, which could have contributed to the onset of Northern Hemisphere glaciation. Although this might in principle be possible, this statement is not corroborated by any quantitative analysis in the paper

and should therefore not be one of the main conclusions in the abstract.

Minor comments

The Antarctic ice sheet boundary conditions are not very clear to me and Figure 1 does not help to clarify this. I would suggest substituting the ice-land-ocean masks in Figure 1 with maps showing the ice sheet topographies for the three different cases.

I cannot find Table S1, which is referenced in the caption of Figure 2.

Dolan, A. M., Haywood, A. M., Hill, D. J., Dowsett, H. J., Hunter, S. J., Lunt, D. J. and Pickering, S. J.: Sensitivity of Pliocene ice sheets to orbital forcing, *Palaeogeogr. Palaeoclimatol. Palaeoecol.*, 309(1-2), 98-110, doi:10.1016/j.palaeo.2011.03.030, 2011.

Lisiecki, L. E. and Raymo, M. E.: Plio-Pleistocene climate evolution: trends and transitions in glacial cycle dynamics, *Quat. Sci. Rev.*, 26(1-2), 56-69, doi:10.1016/j.quascirev.2006.09.005, 2007.

Prescott, C. L., Haywood, A. M., Dolan, A. M., Hunter, S. J., Pope, J. O. and Pickering, S. J.: Assessing orbitally-forced interglacial climate variability during the mid-Pliocene Warm Period, *Earth Planet. Sci. Lett.*, 400, 261-271, doi:10.1016/j.epsl.2014.05.030, 2014.

Willeit, M., Ganopolski, A. and Feulner, G.: On the effect of orbital forcing on mid-Pliocene climate, vegetation and ice sheets, *Clim. Past*, 9(4), 1749-1759, doi:10.5194/cp-9-1749-2013, 2013.

Woodard, S. C., Rosenthal, Y., Miller, K. G., Wright, J. D., Chiu, B. K. and Lawrence, K. T.: Antarctic role in Northern Hemisphere glaciation., *Science*, 847, doi:10.1126/science.1255586, 2014.

Reviewer #2 (Remarks to the Author):

This paper presents a model simulation to explain a proposed shift in Pacific Ocean overturning circulation during the Late Pliocene that ultimately led to onset/intensification of Northern Hemisphere Glaciation. It proposes that this shift related to expansion of marine-based ice sheets in the Pacific sector of West Antarctica. The paper provides modeling support for previous papers that have hypothesized Antarctic drivers that contributed to this climate transition.

A main concern for the paper is that it is very limited in its geological proxy data comparison - restricted largely to the isotope shifts noted by Woodard et al., 2014 (*Science*). It is a very short paper, even by Nature standards. Given this is a Nature Communication paper, the word count and reference list could be doubled. I think this is required before the paper can be accepted to provide a more convincing argument that the processes captured by the model are evident in the geological record.

The paper also states this is a novel explanation for an Antarctic driver for the Plio-Pleistocene transition, but this was proposed by Woodard, and prior to that McKay et al., 2012 (PNAS) - both of which have similar titles (and conclusions) to this paper, and in themselves built on earlier interpretations of Southern Ocean change in sea ice and stratification that may have influenced the carbon cycle. These papers both propose a change in frontal systems and water mass formation (surface, intermediate and deep waters) activity relating to shifting wind fields driven by Antarctic ice advance. While these earlier papers didn't model these changes, the associated changes in proxy records (dust, $\delta^{13}C$) were used to indicate these changes did indeed occur. Woodward noted more efficient heat transport between ocean basins, and this was associated with a quasi-

permanent increase in Antarctic ice volume. For example do the model results capture the observed interoceanic heat transport changes (both deep water and surface) hypothesized in these earlier papers.

However, while the notion of Antarctic drivers for NHG onset is not novel in itself (which should be noted), this manuscript does significantly advance our understanding on some of the potential mechanisms proposed in the earlier papers, because it indicates that the increase in Southern Ocean overturning may predominately have been a Pacific Ocean event related to Pacific variability in the Antarctic Ice Sheets, with implications for the carbon cycle. This makes sense as the majority of ice volume change over this time period would have occurred in this region. What is not clear is what feedbacks are not captured by the coupling of the HadCM3 model to ice volume changes (e.g. changing meltwater inputs, sea ice expansion, changes in the biological pump). While I understand this is a sensitivity test and unrealistic expectation to model, some discussion of these uncaptured feedbacks would greatly benefit the manuscript, in my view.

For example, more discussion on the proposed mechanism for the reduction in CO₂ could be added. At the moment it is very broad (stating only that the reservoir of the deep water mass increased in the Pacific) - can the amount of carbon be quantified in any way?. Is the enhanced Southern Overturning in the model supported by 13C records that indicated a reduction in ventilation of deepwater masses (but increased ventilation of intermediate waters) in the Southern Atlantic Ocean (e.g. Hodell and Venz-Curtis 2006; G3). The methods also state the HadCM3 has a coupled sea ice model - how does sea ice change in response to the ice volume boundary conditions, and what implication does this have on ventilation of the Southern Ocean.

The data and methodological approach seem valid to me and I have no concerns on this front.

Specific comments are:

Line 50-51 - It is stated this has previously been modeled by PlioMIP - so what is novelty of this study. Please clarify point of difference for this paper (I believe it is different from the cited study).

Line 78-79 - do you mean initiation of the ACC? This occurred in the Eocene/Oligocene. Suggest you reword to intensification of the ACC (if this is the current referred to).

Lines 94-97. I found this a bit vague. I suggest the authors are more explicit in how the modeled change in North Pacific storage leads to changes in other carbon reservoirs

Lines 100-102. This sentence is also unclear and contradicts itself. It is not a very strong conclusion- it states changes modeled are reversible, but then maybe not? The model explicitly shows that future ice loss in the Antarctic would be expected to fundamentally alter the southern ocean overturning, and by extension the deep ocean carbon pool. The unknown is the rate at which this would occur in the future.

Reviewer #3 (Remarks to the Author):

General Comments:

A) This study offers an explanation for the empirical observations from Woodard et al. (2014) that show a convergence of $\delta^{18}\text{O}$ and Mg/Ca Temperature from benthic foraminifera at DSDP site 607 in the North Atlantic and ODP site 1208 in the North Pacific. Using the result of modeling work, these authors attribute these changes to a decrease in NPDW due to reduced export of CDW in the absence of sea ice in the Pacific sector of the Antarctic coastline.

B) I find this study intriguing and it has the potential to make a noteworthy and substantive novel contribution by offering a mechanism to explain recent groundbreaking empirical observations of major deep ocean circulation changes at 2.7Ma.

C,E,F) While I think the evidence that NPDW was weaker in the Pliocene is reasonable and this is itself is an novel and important result, at present this study provides insufficient evidence for the mechanism they offer to explain this observation. The text argues that an increase in overturning in the Southern Ocean in response to sea ice expansion leads to greater export of CDW. Figure 3 doesn't clearly make the case for this proposed mechanism. Particularly problematic is Panel E. These figures don't show evidence of an increase in CDW as asserted in the text, rather they show evidence for increase NPDW. Showing the cross section to the margin of Antarctica and having that cross section reveal an increase in CDW seems fairly critical to the credibility of mechanism proposed here.

C,E,F) The link the author's make to CO₂ storage seems tenuous. What is the evidence for the claims that "the export of CDW introduced a water mass that has been isolated from the atmosphere for a long time to large volume of the North Pacific."? As noted above, first there needs to be evidence that export of CDW did in fact increase. Secondly, there needs to be evidence that that water mass had either more carbon or was older. Can they show modeled CO₂ or the modeled age of NPDW between pre-industrial and Pliocene?

D) Table 1, which shows the results of Pliocene MIP simulations using a variety of different models, provides reasonable assurances that the modeling results presented here are not highly sensitive to the model used (i.e. the primary observation of reduced NPDW is reproducible despite different models).

G) This study offers a different mechanism for the empirical datasets of Woodard et al. (2014) than was proposed in the Woodard et al. (2014) paper. The authors should clarify that they are offering a different mechanism in this paper. Woodard et al. (2014) suggest that "Increased Pliocene deep water production in the Pacific-Indian sector of the Southern Ocean potentially allowed AABW to influence the deep Pacific more directly..." This study suggests that less NPDW was formed during the Pliocene not more.

H) Table 1 and the figure caption for figure 3 require more clarity (see below).

Specific Comments:

Lines 28-32: Figure 2 of Woodard et al. 2014 shows the opposite of what the authors suggest here. While the two $\delta^{18}O$ records do converge, it is because the Atlantic record (DSDP 607) is becoming progressively isotopically heavier converging and then matching with the Pacific $\delta^{18}O$ record of ODP 1208 at 2.7 Ma. It is the Pacific $\delta^{18}O$ values that stay within a particular range not the Atlantic. It is the $\delta^{18}O$ of sw records (Figure 3) that show the pattern described by the authors. This distinction is important so the authors should clarify by describing the records more clearly in the text.

Lines 47-49: Again, the transition was towards North Atlantic $\delta^{18}O$ sw not just $\delta^{18}O$ (bf). $\delta^{18}O$ sw is estimated by removing the temperature signature from $\delta^{18}O$ bf. Here the pattern related to $\delta^{18}O$ sw is described, but the authors indicated just $\delta^{18}O$ - which to most paleoceanographers suggests $\delta^{18}O$ bf.

Lines 61-65: Indicate the different panels in Figure 2 that correspond to these simulations? It seems like the first simulation here refers to D in figure 2 while the second refers to C in figure 2. If I am correct here, to prevent confusion C should be the first simulation and D should be the second in figure 2.

Lines 65-66: Be more specific about where the ice shelf is/isn't in a more ice full versus more ice free world?

Lines 74-75: The authors indicate that the change in wind stress on the ocean "changes the most important driver of upwelling and bottom water production in the Southern Ocean" - concisely explaining the "most important driver" process would be helpful here.

Table 1

Explain what + and - indicate in this table.

Figure 3

Panels A-C

It would be helpful to the reader to have what is being show in each of the specific panels more clearly articulated in the caption. I am assuming that the authors are wanting to draw attention to the bright orange regions in panels B and C, but the caption here is confusing here. It says "westerly winds increase in the Pacific sector of the S. Ocean....." Increases relative to what? Are B,C,D plots showing anomalies - i.e. change from no ice in the pacific sector to ice, as implied by "increasing?" Or are they showing the wind, wind stress, and surface mixing conditions for the model run with ice in the Pacific, as implied by the text in lines 70-72?

Panel E

What are the latitudes, units and contours on Panel E? Presumably the figures in this panel span the same latitudes as those in figure 2? It seems fairly important to show these cross sections all the way to the margin of Antarctica to reveal the increase in CDW the authors assert is the cause of augmented NPDW.

We would like to thank the 3 reviewers for all their excellent suggestions. The additions to the manuscript which they have suggested have greatly improved our paper and increased the robustness of our presentation of the results. We have introduced new figures showing the overturning in the Southern Ocean and simulated sea ice and introduced a new section with more direct comparison with the palaeoenvironmental records. Replies to each of the specific comments made by the reviewers are included below.

Reviewers' comments:

Reviewer #1 (Remarks to the Author):

In this paper Hill et al. present results from coupled ocean - atmosphere model simulations of the mid-Pliocene warm period (mPWP) with different configurations of the Antarctic ice sheet. They show that with mPWP boundary conditions an ensemble of models consistently simulate a reduction in the water inflow into the deep North Pacific compared to the present day. Using the HadCM3 model they attribute this change in ocean circulation to the reduced Antarctic ice volume and extent during the mPWP. The authors then describe the possible implications of this circulation change for the onset of Northern Hemisphere glaciation during the Pliocene-Pleistocene transition.

As acknowledged by the authors, the idea that an increase in the Antarctic ice volume could have affected the ocean overturning circulation has been already put forward by (Woodard et al., 2014) based on temperature and isotope reconstructions from two ODP sites in the North Atlantic and North Pacific. This paper adds a quantitative estimate of the possible contribution of changes in Antarctica to the overturning ocean circulation and describes possible mechanisms responsible for the circulation changes. The paper is definitely of interest for readers interested in the mechanisms that caused the Pliocene-Pleistocene transition. I therefore recommend publication if the authors can satisfactorily address some major comments outlined below.

Main comments

The results presented show a consistent weakening of the Pacific overturning circulation in idealized time-slab mPWP model simulations compared to present day. However, simulations of a single model are used to attribute the changes in Pacific circulation to the state of the Antarctic ice sheet. Could the authors elaborate on how representative the results are for the rest of the model ensemble? Without the need to run additional simulations, could the authors include some analysis of whether the pattern of changes in wind stress and mixed layer depth are a robust feature among the PlioMIP model ensemble or not?

New analysis of the PlioMIP ensemble has been added to the manuscript, through an extra column in the PlioMIP table. This details changes in the simulations of winds over the Pacific sector of the Southern Ocean. The ensemble data does not include wind stress or the mixed layer depth, but as these are closely related to the wind strength we feel that these results are sufficient to show that the other models in the ensemble are behaving in a similar manner in the Pacific to HadCM3. These data significantly strengthen our argument that the PlioMIP ensemble is behaving in a similar way.

The paper concentrates on the modeled differences in overturning circulation for an average 'interglacial' during the mPWP. However, both the mPWP and the subsequent Plio-Pleistocene transition are characterized by substantial orbital scale variability of both climate and ice sheets, particularly at the 40 kyr time scale corresponding to obliquity variations (Dolan et al., 2011; Lisiecki and Raymo, 2007; Prescott et al., 2014; Willeit et al., 2013). Thus the question arises whether the circulation changes described in the paper for the interglacials persisted also during glacial periods. Could the authors comment on that?

We have added a few sentences discussing G-IG cycles in both the Pliocene and Pleistocene and the possible role of the presented mechanisms in these climate changes. In general this analysis is beyond the scope of the current manuscript, but it does suggest an interesting follow up study.

The last part of the abstract (lines 17-21) includes strong statements without any quantitative estimate of the effect of a circulation change on the mentioned processes:

"This explains why the isotope records from the Pacific and Atlantic Oceans converge at the Plio-Pleistocene transition and suggests a novel explanation of the geologically rapid decrease in atmospheric carbon dioxide concentrations at the Plio-Pleistocene transition, the stratification of the North Pacific and the initiation of Northern Hemisphere glaciation." In particular I'm missing a quantitative estimate of the changes in deep North Atlantic and North Pacific temperatures induced by the circulation change. These could easily be compared to the reconstructed temperatures presented in (Woodard et al., 2014) and could give some support to the claim made by the authors that it was this change in overturning circulation that was responsible for the convergence of bottom water temperatures and (possibly) the isotopic records of the North Atlantic and Pacific.

Although the model does not produce simulation of ocean isotope ratios, we have added a comparison of ocean temperatures at the sites used in the Woodard et al. 2014 paper. This is incorporated into a new table, detailing the ocean temperatures in each of the four HadCM3 simulations used in this study.

The authors also argue that the increase in deep water inflow into the North Pacific could have increased the carbon stored in the ocean thereby causing the atmospheric CO₂ to drop, which could have contributed to the onset of Northern Hemisphere glaciation. Although this might in principle be possible, this statement is not corroborated by any quantitative analysis in the paper and should therefore not be one of the main conclusions in the abstract.

The abstract has been reworded to show that this is only a suggested further consequence of the results presented here, rather than something that this work quantifies. The current class of models used for simulating the Pliocene (both our own and those within PlioMIP) do not include the carbon cycle. Now we have shown changes in the Pacific Ocean circulation during the Pliocene, this is definitely something we would like to follow up, although it requires a different class of model.

Minor comments

The Antarctic ice sheet boundary conditions are not very clear to me and Figure 1 does not help to

clarify this. I would suggest substituting the ice-land-ocean masks in Figure 1 with maps showing the ice sheet topographies for the three different cases.

These topographies are available elsewhere and could be easily plotted for a replacement figure. However, the extent of the ice sheet would not be nearly as clear and as such we suggest that the current figure is more appropriate and helpful for the reader to understand the mechanism being presented here. However, we have added a reference to the figure caption in which the global orography is presented for the HadCM3 PlioMIP and pre-industrial simulations.

I cannot find Table S1, which is referenced in the caption of Figure 2.

Error corrected.

Dolan, A. M., Haywood, A. M., Hill, D. J., Dowsett, H. J., Hunter, S. J., Lunt, D. J. and Pickering, S. J.: Sensitivity of Pliocene ice sheets to orbital forcing, *Palaeogeogr. Palaeoclimatol. Palaeoecol.*, 309(1-2), 98-110, doi:10.1016/j.palaeo.2011.03.030, 2011.

Lisiecki, L. E. and Raymo, M. E.: Plio-Pleistocene climate evolution: trends and transitions in glacial cycle dynamics, *Quat. Sci. Rev.*, 26(1-2), 56-69, doi:10.1016/j.quascirev.2006.09.005, 2007.

Prescott, C. L., Haywood, A. M., Dolan, A. M., Hunter, S. J., Pope, J. O. and Pickering, S. J.: Assessing orbitally-forced interglacial climate variability during the mid-Pliocene Warm Period, *Earth Planet. Sci. Lett.*, 400, 261-271, doi:10.1016/j.epsl.2014.05.030, 2014.

Willeit, M., Ganopolski, A. and Feulner, G.: On the effect of orbital forcing on mid-Pliocene climate, vegetation and ice sheets, *Clim. Past*, 9(4), 1749-1759, doi:10.5194/cp-9-1749-2013, 2013.

Woodard, S. C., Rosenthal, Y., Miller, K. G., Wright, J. D., Chiu, B. K. and Lawrence, K. T.: Antarctic role in Northern Hemisphere glaciation., *Science*, 847, doi:10.1126/science.1255586, 2014.

Reviewer #2 (Remarks to the Author):

This paper presents a model simulation to explain a proposed shift in Pacific Ocean overturning circulation during the Late Pliocene that ultimately led to onset/intensification of Northern Hemisphere Glaciation. It proposes that this shift related to expansion of marine-based ice sheets in the Pacific sector of West Antarctica. The paper provides modeling support for previous papers that have hypothesized Antarctic drivers that contributed to this climate transition.

A main concern for the paper is that it is very limited in its geological proxy data comparison - restricted largely to the isotope shifts noted by Woodard et al., 2014 (*Science*). It is a very short paper, even by Nature standards. Given this is a Nature Communication paper, the word count and reference list could be doubled. I think this is required before the paper can be accepted to provide a

more convincing argument that the processes captured by the model are evident in the geological record.

The paper also states this is a novel explanation for an Antarctic driver for the Plio-Pleistocene transition, but this was proposed by Woodard, and prior to that McKay et al., 2012 (PNAS) - both of which have similar titles (and conclusions) to this paper, and in themselves built on earlier interpretations of Southern Ocean change in sea ice and stratification that may have influenced the carbon cycle. These papers both propose a change in frontal systems and water mass formation (surface, intermediate and deep waters) activity relating to shifting wind fields driven by Antarctic ice advance. While these earlier papers didn't model these changes, the associated changes in proxy records (dust, $\delta^{13}C$) were used to indicate these changes did indeed occur. Woodward noted more efficient heat transport between ocean basins, and this was associated with a quasi-permanent increase in Antarctic ice volume. For example do the model results capture the observed interoceanic heat transport changes (both deep water and surface) hypothesized in these earlier papers.

We have added a significant element of data-model comparison to the paper. The Pacific carbon isotope changes between the Pliocene and Pleistocene (Ravelo and Andreason, 2000) are qualitatively compared to the proposed ocean circulation changes. Direct comparison to the deep ocean temperature changes seen in the North Pacific and North Atlantic (Woodard et al., 2014) have been added in Table 2. Comparison has also been made with the ANDRILL record from the Pacific-sector of the Antarctic (McKay et al., 2012). All of these have been incorporated into a new data-model comparison section of the results.

However, while the notion of Antarctic drivers for NHG onset is not novel in itself (which should be noted), this manuscript does significantly advance our understanding on some of the potential mechanisms proposed in the earlier papers, because it indicates that the increase in Southern Ocean overturning may predominately have been a Pacific Ocean event related to Pacific variability in the Antarctic Ice Sheets, with implications for the carbon cycle. This makes sense as the majority of ice volume change over this time period would have occurred in this region. What is not clear is what feedbacks are not captured by the coupling of the HadCM3 model to ice volume changes (e.g. changing meltwater inputs, sea ice expansion, changes in the biological pump). While I understand this is a sensitivity test and unrealistic expectation to model, some discussion of these uncaptured feedbacks would greatly benefit the manuscript, in my view.

We already had some discussion of uncaptured feedbacks in relation to the carbon cycle and have added these glaciological factors to that section of the discussion. We have added a new figure detailing the changes in Southern Hemisphere sea ice in these simulations.

For example, more discussion on the proposed mechanism for the reduction in CO_2 could be added. At the moment it is very broad (stating only that the reservoir of the deep water mass increased in the Pacific) - can the amount of carbon be quantified in any way?. Is the enhanced Southern Overturning in the model supported by ^{13}C records that indicated a reduction in ventilation of deepwater masses (but increased ventilation of intermediate waters) in the Southern Atlantic Ocean (e.g. Hodell and Venz-Curtis 2006; G3). The methods also state the HadCM3 has a coupled sea ice

model - how does sea ice change in response to the ice volume boundary conditions, and what implication does this have on ventilation of the Southern Ocean.

We have included a section entitled Comparisons to Plio-Pleistocene records from the Pacific Ocean, which incorporates qualitative discussion of how the $\delta^{13}C$ records compare well with the changes seen in the simulations. Hodell and Venz-Curtis (2006) do not include sites from the Southern Ocean and only 1 site from the Pacific Ocean (849). As this site also appears in Ravelo and Andreason (2000), along with 3 other sites from the Pacific, we use this data set for comparison to the model.

We have added a figure (Fig. 4) detailing the Southern Hemisphere sea ice simulated in HadCM3. We note a non-linear geographical response to terrestrial ice extent. Thus, any impact of ice extent on Southern Ocean or deep water ventilation would need to be modelled to determine both the magnitude and sign of any change.

The data and methodological approach seem valid to me and I have no concerns on this front.

Specific comments are:

Line 50-51 - It is stated this has previously been modeled by PlioMIP - so what is novelty of this study. Please clarify point of difference for this paper (i believe it is different from the cited study).

One of the HadCM3 simulations presented here is the same as the PlioMIP simulations (PlioMIP standard), although Pacific circulation has never been looked at in these simulations. This section has been rewritten to make it clearer, that there is a PlioMIP simulation and then 2 simulations with changes in Antarctica.

Line 78-79 - do you mean initiation of the ACC? This occurred in the Eocene/Oligocene. Suggest you reword to intensification of the ACC (if this is the current referred to).

We were referring to NPDW, have reworded to make this clear.

Lines 94-97. I found this a bit vague. I suggest the authors are more explicit in how the modeled change in North Pacific storage leads to changes in other carbon reservoirs

We were referring to possible feedbacks, as the other carbon pools respond to the cooling associated with increased ocean carbon storage. Have edited the text to reflect this.

Lines 100-102. This sentence is also unclear and contradicts itself. It is not a very strong conclusion- it states changes modeled are reversible, but then maybe not? The model explicitly shows that future ice loss in the Antarctic would be expected to fundamentally alter the southern ocean overturning, and by extension the deep ocean carbon pool. The unknown is the rate at which this would occur in the future.

We have not done the same experiments for either the pre-industrial or future climate change scenarios, so we cannot say that these simulations definitely show that these changes would occur

in future. However, the reversibility of our results hints that this could be the case. Have added to the final sentences to clear up this confusion.

Reviewer #3 (Remarks to the Author):

General Comments:

A) This study offers an explanation for the empirical observations from Woodard et al. (2014) that show a convergence of $\delta^{18}\text{O}$ and Mg/Ca Temperature from benthic foraminifera at DSDP site 607 in the North Atlantic and ODP site 1208 in the North Pacific. Using the result of modeling work, these authors attribute these changes to a decrease in NPDW due to reduced export of CDW in the absence of sea ice in the Pacific sector of the Antarctic coastline.

B) I find this study intriguing and it has the potential to make a noteworthy and substantive novel contribution by offering a mechanism to explain recent groundbreaking empirical observations of major deep ocean circulation changes at 2.7Ma.

C,E,F) While I think the evidence that NPDW was weaker in the Pliocene is reasonable and this is itself is an novel and important result, at present this study provides insufficient evidence for the mechanism they offer to explain this observation. The text argues that an increase in overturning in the Southern Ocean in response to sea ice expansion leads to greater export of CDW. Figure 3 doesn't clearly make the case for this proposed mechanism. Particularly problematic is Panel E. These figures don't show evidence of an increase in CDW as asserted in the text, rather they show evidence for increase NPDW. Showing the cross section to the margin of Antarctica and having that cross section reveal an increase in CDW seems fairly critical to the credibility of mechanism proposed here.

We have added a figure showing the changes in the CDW. These support our previous statements about the mechanisms in the model causes changes to the NPDW. In response to the previous reviewer we have also added in details of the changes in Pacific sector Southern Ocean wind fields from the other PlioMIP simulations, which show that similar things are happening in these simulations.

C,E,F) The link the author's make to CO₂ storage seems tenuous. What is the evidence for the claims that "the export of CDW introduced a water mass that has been isolated from the atmosphere for a long time to large volume of the North Pacific."? As noted above, first there needs to be evidence that export of CDW did in fact increase. Secondly, there needs to be evidence that that water mass had either more carbon or was older. Can they show modeled CO₂ or the modeled age of NPDW between pre-industrial and PlioMIP?

GCMs do not usually model CO₂ or the age of waters, so we are unable to show these values. However, we do show that in the Pliocene most of the Pacific deep waters return to the source of ventilation in the Southern Ocean without entering the North Pacific and therefore feel that it is

justified to talk about it being “a water mass isolated from the atmosphere”. Now that we have shown significant differences in Pliocene Pacific Ocean circulation the use of a coupled climate - carbon cycle model would be an interesting further study.

D) Table 1, which shows the results of Pliocene MIP simulations using a variety of different models, provides reasonable assurances that the modeling results presented here are not highly sensitive to the model used (i.e. the primary observation of reduced NPDW is reproducible despite different models).

G) This study offers a different mechanism for the empirical datasets of Woodard et al. (2014) than was proposed in the Woodard et al. (2014) paper. The authors should clarify that they are offering a different mechanism in this paper. Woodard et al. (2014) suggest that "Increased Pliocene deep water production in the Pacific-Indian sector of the Southern Ocean potentially allowed AABW to influence the deep Pacific more directly..." This study suggests that less NPDW was formed during the Pliocene not more.

We have added a sentence to the Pliocene Southern Ocean results making the differences in the simulated mechanism and that suggested by Woodard et al. clear.

H) Table 1 and the figure caption for figure 3 require more clarity (see below).

Specific Comments:

Lines 28-32: Figure 2 of Woodard et al. 2014 shows the opposite of what the authors suggest here. While the two d18O records do converge, it is because the Atlantic record (DSDP 607) is becoming progressively isotopically heavier converging and then matching with the Pacific d18O record of ODP 1208 at 2.7 Ma. It is the Pacific d18O values that stay within a particular range not the Atlantic. It is the d18O of sw records (Figure 3) that show the pattern described by the authors. This distinction is important so the authors should clarify by describing the records more clearly in the text.

Everywhere where the North Pacific and North Atlantic d18O is referenced within the text, we have made it clear in the text that we are referring to the sea water oxygen isotopes present by Woodard et al., 2014.

Lines 47-49: Again, the transition was towards North Atlantic d18O sw not just d18O (bf). d18Osw is estimated by removing the temperature signature from d18O bf. Here the pattern related to d18O sw is described, but the authors indicated just d18O - which to most paleoceanographers suggests d18O bf.

We have added “sea water” to the second mention of oxygen isotopes.

Lines 61-65: Indicate the different panels in Figure 2 that correspond to these simulations? It seems like the first simulation here refers to D in figure 2 while the second refers to C in figure 2. If I am correct here, to prevent confusion C should be the first simulation and D should be the second in figure 2.

We have added panels to references to Figure 2.

Lines 65-66: Be more specific about where the ice shelf is/isn't in a more ice full versus more ice free world?

We have added in reference to the specific areas of Pacific sector of Antarctica in which the PRISM boundary conditions remove ice.

Lines 74-75: The authors indicate that the change in wind stress on the ocean "changes the most important driver of upwelling and bottom water production in the Southern Ocean" - concisely explaining the "most important driver" process would be helpful here.

Table 1

Explain what + and - indicate in this table.

We have changed this so that it represents the change at the as we move from the Pliocene into the Pleistocene and added description of the precise nature of the anomalies to the caption.

Figure 3

Panels A-C

It would be helpful to the reader to have what is being show in each of the specific panels more clearly articulated in the caption. I am assuming that the authors are wanting to draw attention to the bright orange regions in panels B and C, but the caption here is confusing here. It says "westerly winds increase in the Pacific sector of the S. Ocean....." Increases relative to what? Are B,C,D plots showing anomalies - i.e. change from no ice in the pacific sector to ice, as implied by "increasing?" Or are they showing the wind, wind stress, and surface mixing conditions for the model run with ice in the Pacific, as implied by the text in lines 70-72?

We have made the anomalies shown explicit in the figure caption.

Panel E

What are the latitudes, units and contours on Panel E? Presumably the figures in this panel span the same latitudes as those in figure 2? It seems fairly important to show these cross sections all the way to the margin of Antarctica to reveal the increase in CDW the authors assert is the cause of augmented NPDW.

We have removed panels A and E, as the images are already shown in Figs 1 and 2 and have added a figure (Fig. 3) showing CDW changes in the simulations.

Reviewers' comments:

Reviewer #1 (Remarks to the Author):

In this revised version of their manuscript, the authors have satisfactorily addressed and responded to my comments and included some additional analyses as suggested. I therefore recommend publication of the revised manuscript. I only have some minor technical comments:

- Table 2: units should be given in the table or the caption.
- Figure 3: If I understand correctly, the results from Pliomip simulations in panels B-D are shown as Pliomip - Preindustrial. If this is correct and AABW and CDW are weaker in the Pliocene than in the Preindustrial, then there is probably something wrong with the colorbar below panel B.

Reviewer #2 (Remarks to the Author):

I have assessed the revisions and I am satisfied the authors have addressed my concerns, which largely involved an expansion of the discussion on how the model results compare with proxy records. I like the new discussion around AABW formation and the links to Antarctic/deep sea records - and although it contradicts the Woodard interpretation of this change, intuitively this makes more sense to me and is supported by the proximal proxy data (both Antarctic and deep sea - as noted in lines 148-156). While ANDRILL supports this for the Ross Sea, the new sea ice model figure (fig 4) suggests these changes also occurred in the other key bottom water formation area, most notably Adelie Land Bottom water formation area in the PlioMIP standard run. The addition of Figure 3 is also excellent.

I recommend acceptance without the need for further revision

Reviewer #3 (Remarks to the Author):

Major Comments:

The tables and figures in this revision are insufficiently explained, which significantly undermines the reader's ability to evaluate the evidence for the core argument advanced in this paper (see below for details). For example, evaluating the evidence for the assertions in lines 90-94 isn't feasible without a clearer explanation of figure 3. The interpretations need to be more clearly connected to specific results by the authors explaining what is presented in the figures and how that information informs their interpretations.

I still find the link made to an increase in CO₂ storage tenuous. If the authors' interpretation of figure 3 (see comments below) is sound then the first condition of evidence, that the export of CDW increased, was met here. However, the evidence for a clear link between the circulation changes modeled here and the global carbon cycle is not met. Thus, I still find the discussion section here is too speculative. In the absence of evidence for a clear link, framing the argument for greater CO₂ storage as a hypothesis and articulating how that hypothesis could be tested by making specific predictions about what should be observed via either empirical data or from modeling work would be much more defensible than the current approach. This recommendation extends to the abstract where the potential impact on the carbon cycle is also referenced.

Some of the responses to reviews incorporated in the text doesn't address the specific query fully and in some places disrupts the flow of the paper. For example in the section on comparisons to Plio-Pleistocene records the authors insert data about carbon isotopes in response to a reviewer

comment, but then don't explain those data very clearly and don't make the connection to their model results or associated interpretation in a compelling way. I find this revision less compelling than the initial manuscript.

Specific Comments:

Line 20: The last line of the abstract seems to be missing a preposition: "This mechanism may also cause atmospheric carbon dioxide [to] decrease,..."

Lines 28-30: It isn't just seawater $\delta^{18}\text{O}$ that converges at this time. See Woodard et al. 2014 figures 2 and 3. $\delta^{18}\text{O}$ calcite, Mg/Ca BWT and $\delta^{18}\text{O}_{\text{sw}}$ all converge at this time.

Lines 30-31 you mean here specifically $\delta^{18}\text{O}_{\text{sw}}$. As written this statement could potentially mean other isotopic systems (e.g. D/H).

Line 34: "suggesting that the North Pacific no longer responds to local forcing" specify that this clause is referencing the time after the transition.

Lines 77-78 Not all readers will know where these regions are. Indicate the location of these geographic regions on a figure. Perhaps on figure 1?

Lines 87-90: Figure 3A is not a good illustration of the text here in terms of demarcating AABW or showing its flow.

Lines 100-101: This sentence is vague. What is meant by "other changes in the Pliocene Earth System"?

Lines 106-108 explain how these assertions are validated by the panels shown in figure 4.

Line 110 -112 show where the Bellingshausen Sea and Ross Sea are on figure 1. However, even if the localities were labeled, these assertions are not compellingly advanced by panels C and D in Figure 4.

Lines 129-146 This section seeks to connect the results of empirical studies to the model results in this study. But again explicit ties between the data and model results that lead to the articulated interpretation are missing (E.g. how do these carbon isotopes observations corroborate the modeling work) which makes this section not very compelling.

Lines 152 How do increases in diatoms that favour polar open ocean or seasonal sea ice provide evidence of an intensification of Antarctic coastal wind fields?

Lines 182-186: Its not clear what these sentences add to the paper. What are the other carbon reservoirs? Why reference G-IG changes in CO_2 related to N Pacific process here? How do these references specifically relate to the core argument advanced in this paper?

Table 1: Again, what do + and - mean in this table. Be explicit.

Table 2: what are the units here? Presumably $^{\circ}\text{C}$?

Figures 2 and 3: I don't see any indication in either caption of what + and - Sv means in terms of flow direction. While I think this is an issue that needs to be addressed for both figures, it is significantly more problematic for interpreting Figure 3 (see below).

Figure 3 :I am underwhelmed by AABW as a watermass in panel A. What here exactly is AABW? The weak negative contours near the surface that are furthest south? There is an arrow labeled

AABW that spans 40° of latitude and much of which corresponds to 0 sverdrups. In the absence of clarity about what constitutes AABW evaluating whether it has strengthened or weakened in the subsequent panels isn't feasible.

Figure 3: Clarify what is meant by "relative to the pre-industrial experiment" in panels 3B-D. Again what does + and - Sv mean? How is it that negative and positive Sv can both be indicating a weakening/reduction as is suggested in the caption? "The standard Pliocene simulation (B) shows that CDW is weaker and that AABW formation and export is significantly reduced."

Figure 3: Note that there is also no Y axis label in this figure. Presumably the units are meters. Labeling the X axis "latitude" also wouldn't hurt.

Figure 5. How were these panels generated? The caption says "modelled mechanism by which Antarctic ice advance changes ocean circulation." What were the boundary conditions here? Are they for one of the simulations? The caption says westerly winds increase - relative to what baseline? A clearer explanation of what is being depicted here is required in order for lines 114 to 127 to be evaluated.

Figure 5. What do + and - values in each panel mean in terms of flow direction? Where are the Westerly winds here? Most readers can infer these things, but making a compelling case means that what is displayed and how it related to the arguments being made should be very clear.

Reviewer #3 (Remarks to the Author):

Major Comments:

The tables and figures in this revision are insufficiently explained, which significantly undermines the reader's ability to evaluate the evidence for the core argument advanced in this paper (see below for details). For example, evaluating the evidence for the assertions in lines 90-94 isn't feasible without a clearer explanation of figure 3. The interpretations need to be more clearly connected to specific results by the authors explaining what is presented in the figures and how that information informs their interpretations.

I still find the link made to an increase in CO₂ storage tenuous. If the authors' interpretation of figure 3 (see comments below) is sound then the first condition of evidence, that the export of CDW increased, was met here. However, the evidence for a clear link between the circulation changes modeled here and the global carbon cycle is not met. Thus, I still find the discussion section here is too speculative. In the absence of evidence for a clear link, framing the argument for greater CO₂ storage as a hypothesis and articulating how that hypothesis could be tested by making specific predictions about what should be observed via either empirical data or from modeling work would be much more defensible than the current approach. This recommendation extends to the abstract where the potential impact on the carbon cycle is also referenced.

Using the models and techniques of this study cannot quantify the changes to the carbon cycle. It was always our intention to frame the increase in carbon storage as a hypothesis. As such, the reference to this has been removed from the abstract and the discussion has been changed in order to make this clear. We have presented our new results as identifying a novel change in the ocean that could have an important impact on the carbon cycle. The next stage of this work would be to simulate the Pliocene carbon cycle and quantify the effects of ocean changes on atmospheric carbon dioxide concentrations.

Some of the responses to reviews incorporated in the text doesn't address the specific query fully and in some places disrupts the flow of the paper. For example in the section on comparisons to Plio-Pleistocene records the authors insert data about carbon isotopes in response to a reviewer comment, but then don't explain those data very clearly and don't make the connection to their model results or associated interpretation in a compelling way. I find this revision less compelling than the initial manuscript.

Specific Comments:

Line 20: The last line of the abstract seems to be missing a preposition: "This mechanism may also cause atmospheric carbon dioxide [to] decrease,..."

This line has been removed.

Lines 28-30: It isn't just seawater d¹⁸O that converges at this time. See Woodard et al. 2014 figures 2 and 3. D¹⁸O calcite, Mg/Ca BWT and d¹⁸O_{sw} all converge at this time.

Added in reference to the oxygen isotopes of calcite and bottom water temperatures converging at this time.

Lines 30-31 you mean here specifically $\delta^{18}\text{O}_{\text{sw}}$. As written this statement could potentially mean other isotopic systems (e.g. D/H).

Added word oxygen to make this clear.

Line 34: "suggesting that the North Pacific no longer responds to local forcing" specify that this clause is referencing the time after the transition.

Added a clause to the sentence to make this clear.

Lines 77-78 Not all readers will know where these regions are. Indicate the location of these geographic regions on a figure. Perhaps on figure 1?

We have referenced papers detailing these regions and made the locations more clear.

Lines 87-90: Figure 3A is not a good illustration of the text here in terms of demarcating AABW or showing its flow.

Southern Ocean overturning drives the formation of AABW and its export into the world's deep oceans. Admittedly this plot does not make the main flow of AABW down the continental shelf clear, as this occurs at only a few locations and is largely overwhelmed when integrated over all longitudes. However, the inclusion of the typical flow path of the AABW helps the reader to understand the discussion in the text about the changes in the overturning.

Lines 100-101: This sentence is vague. What is meant by "other changes in the Pliocene Earth System"?

Made it clear that we were referring to the other boundary condition changes in the model and referenced the PlioMIP experimental design.

Lines 106-108 explain how these assertions are validated by the panels shown in figure 4.

PlioMIP w. mod Antarctica shows a sea ice extent intermediate between the Pre-industrial and the standard Pliocene simulation.

Line 110 -112 show where the Bellingshausen Sea and Ross Sea are on figure 1. However, even if the localities were labeled, these assertions are not compellingly advanced by panels C and D in Figure 4.

Have removed the example.

Lines 129-146 This section seeks to connect the results of empirical studies to the model results in this study. But again explicit ties between the data and model results that lead to the articulated interpretation are missing (E.g. how do these carbon isotopes observations corroborate the modeling work) which makes this section not very compelling.

As we point out in the manuscript the data and models cannot be explicitly compared, as climate models do not typically run with carbon isotopes. This is further work that could be done in the future, but running such simulations with a single model would certainly be a manuscript in its own

right. With those caveats, we do feel that the available data reflects what would be anticipated from the ocean circulation changes presented here. The comparison is certainly not compelling in itself, but conversely does not present results at odds with our simulations.

Lines 152 How do increases in diatoms that favour polar open ocean or seasonal sea ice provide evidence of an intensification of Antarctic coastal wind fields?

Antarctic polynyas are caused by coastal wind fields and would provide the habitat for open ocean diatoms close to the Antarctic coast.

Lines 182-186: Its not clear what these sentences add to the paper. What are the other carbon reservoirs? Why reference G-IG changes in CO₂ related to N Pacific process here? How do these references specifically relate to the core argument advanced in this paper?

As we cannot constrain the magnitude of the effect on the carbon stored in the deep Pacific using this particular modelling framework and we know there have been changes in other carbon stores (although ones with lower potential for carbon storage change) it seemed appropriate to mention these. Similar, during G-IG cycles lots of carbon stores changes, but the North Pacific does seem to play an important role. This seemed like important information to convey to the reader.

Table 1: Again, what do + and - mean in this table. Be explicit.

Added the word "negative" to make clear to the reader that anti-clockwise circulation is negative in overturning calculations.

Table 2: what are the units here? Presumably °C?

Added units to table.

Figures 2 and 3: I don't see any indication in either caption of what + and - Sv means in terms of flow direction. While I think this is an issue that needs to be addressed for both figures, it is significantly more problematic for interpreting Figure 3 (see below).

Added a sentence explaining negative and positive values on MOC plots.

Figure 3 :I am underwhelmed by AABW as a watermass in panel A. What here exactly is AABW? The weak negative contours near the surface that are furthest south? There is an arrow labeled AABW that spans 40° of latitude and much of which corresponds to 0 sverdrups. In the absence of clarity about what constitutes AABW evaluating whether it has strengthened or weakened in the subsequent panels isn't feasible.

AABW is hard to represent on simple diagrams, being formed in several localities and latitudes and taking a number of complex pathways into the deep ocean. Representing it by the overturning circulation has weaknesses, but is one of the few ways of showing what is going on in the Southern Ocean in a coherent way. The AABW has significant flows down the Antarctic continental slope, but this occurs at a few locations and when summed across the globe and represented by MOC, this can lead to the signal being overwhelmed. However, by plotting this you can see the mean surface overturning representing the formation of the AABW and the deep overturning that exports it to the global oceans. Seeing these important processes and there changes between the simulations on the plot is more important than representing the flow down the continental shelf.

Figure 3: Clarify what is meant by "relative to the pre-industrial experiment" in panels 3B-D. Again what does + and - Sv mean? How is it that negative and positive Sv can both be indicating a weakening/reduction as is suggested in the caption? "The standard PlioMIP simulation (B) shows that CDW is weaker and that AABW formation and export is significantly reduced."

Added a sentence explaining negative and positive values on MOC plots.

Figure 3: Note that there is also no Y axis label in this figure. Presumably the units are meters. Labeling the X axis "latitude" also wouldn't hurt.

Added labels for axes "m.b.s.l." and "Latitude" and an explanation of the m.b.s.l. to the figure caption.

Figure 5. How were these panels generated? The caption says "modelled mechanism by which Antarctic ice advance changes ocean circulation." What were the boundary conditions here? Are they for one of the simulations? The caption says westerly winds increase - relative to what baseline? A clearer explanation of what is being depicted here is required in order for lines 114 to 127 to be evaluated.

Have moved the sentence that describes exactly which experiments are being compared to the front of the caption, so that the reader will be clear before examining each panel what is being plotted.

Figure 5. What do + and - values in each panel mean in terms of flow direction? Where are the Westerly winds here? Most readers can infer these things, but making a compelling case means that what is displayed and how it related to the arguments being made should be very clear.

The way it was written was a little ambiguous, so have added a clear demarcation of the meaning of positive and negative values.

REVIEWERS' COMMENTS:

Reviewer #3 (Remarks to the Author):

General comments:

Overall this revision is a significant improvement from the last revision. I still think there are a number of places where the authors need to clarify their assertions and expound upon what they mean, which I have recorded in the specific comments below. I think if these concerns are addressed to the satisfaction of the editor, then this paper is publication worthy.

Specific comments:

Line 18: I think this statement should be more guarded - replace "this explains" with "these results potentially explain"

Lines 29-30: the phrasing is awkward here.

Line 37: clarify what "this" is here.

Line 91-93: I think the authors have provided a reasonable argument in their rebuttal about why AABW is underwhelming in Fig 3A, but I think that argument needs to appear somewhere in the manuscript, either here or in the figure caption for Figure 3.

Lines 103: Modify this sentence so it informs the reader about what Woodard et al invoke greater deep water formation to explain. Also, the mechanism invoked by Woodard et al is derived from modeling results of Zhang et al. 2013 (Nat Com). That paper probably also deserves a citation here.

Lines 109-111: Figure 4 doesn't show anything about how "general warming during the mPWP" impacts sea ice around Antarctica.

Lines 135-138: I appreciate this reframing of the C isotopes. However, I think one or two more sentences are need here (i.e. just after "would be expected from the simulations presented here" explain what the predictions are based upon the models and how the existing carbon isotope data are consistent with that prediction).

Lines 167-169 "This previously unknown effect on the ocean carbon storage potential..." more clarity is required here. what previously unknown effect are the authors talking about?

Lines 171-174: This change provides a plausible mechanisms for Antarctica to drive the Plio-Pleistocene transition...." In order for Ice to expand in Antarctica there needs to have been a climatic change that lead to the expansion of ice. So it seems erroneous to invoke Antarctic ice expansion as the ultimate cause of the Plio-Pleistocene transition and the intensification of NHG.

Lines 193-194: I am confused by the use of "however" to start this sentence. Doesn't the sentence before it say that the N Pacific may play a big role on long timescales. This statement seems to say the N Pacific also plays a role in G-IG timescales. I recommend omitting the "However" here.

Lines 197-199: I appreciate that the authors have toned down the carbon cycle part of this paper. Here, I happen to think this last statement doesn't have to be as guarded as it is. It would be fair to say that the changes in deep ocean circulation could potentially have significant impacts on the deep ocean carbon cycle.

Figure 3 Caption: "The Pliocene simulations are show relative to the pre-industrial experiment."

Which direction was the difference taken? Plio-Pre-industrial or visa versa?

Figure 3: I appreciate that the authors have now indicated what the negative and positive values mean in MOC plots. But I would submit that if the authors want this figure to be meaningful to any non-modeler reading it they need to provide more help to the reader to understand how to interpret the anomaly panels. Seeing more contour lines on a plot generally indicates more not less. Explaining more clearly that in these plots more deeply contoured features (both postive and negative) mean a reduction not an increase in watermass strength because they are anomalies would be helpful.

Figure 5: Include arrows that indicate ACC and Antarctic counter current?

REVIEWERS' COMMENTS:

Reviewer #3 (Remarks to the Author):

General comments:

Overall this revision is a significant improvement from the last revision. I still think there are a number of places where the authors need to clarify their assertions and expound upon what they mean, which I have recorded in the specific comments below. I think if these concerns are addressed to the satisfaction of the editor, then this paper is publication worthy.

Specific comments:

Line 18: I think this statement should be more guarded - replace "this explains" with "these results potentially explain"

Replaced the text with the suggested statement.

Lines 29-30: the phrasing is awkward here.

Have split the sentence into two sentences, so it is less awkward.

Line 37: clarify what "this" is here.

Have reworded the sentence.

Line 91-93: I think the authors have provided a reasonable argument in their rebuttal about why AABW is underwhelming in Fig 3A, but I think that argument needs to appear somewhere in the manuscript, either here or in the figure caption for Figure 3.

Have added a sentence into the text outlining the basic argument, which will hopefully also help the reader interpret Figure 3.

Lines 103: Modify this sentence so it informs the reader about what Woodard et al invoke greater deep water formation to explain. Also, the mechanism invoked by Woodard et al is derived from modeling results of Zhang et al. 2013 (Nat Com). That paper probably also deserves a citation here.

Expanded the sentence as suggested and incorporated the reference.

Lines 109-111: Figure 4 doesn't show anything about how "general warming during the mPWP" impacts sea ice around Antarctica.

Have changed the phrasing of the sentence, so it is clearer.

Lines 135-138: I appreciate this reframing of the C isotopes. However, I think one or two more sentences are need here (i.e. just after "would be expected from the simulations presented here" explain what the predictions are based upon the models and how the existing carbon isotope data are consistent with that prediction).

Have added a sentence at the end of the section on C isotopes to make clear what aspects of simulation are in accord with the measurements.

Lines 167-169 "This previously unknown effect on the ocean carbon storage potential..." more clarity is required here. what previously unknown effect are the authors talking about?

Have added some words to the sentence to make this clear.

Lines 171-174: This change provides a plausible mechanisms for Antarctica to drive the Plio-Pleistocene transition....." In order for Ice to expand in Antarctica there needs to have been a climatic change that lead to the expansion of ice. So it seems erroneous to invoke Antarctic ice expansion as the ultimate cause of the Plio-Pleistocene transition and the intensification of NHG.

Made it clear that this referring to the oceans, which is what we have modelled.

Lines 193-194: I am confused by the use of "however" to start this sentence. Doesn't the sentence before it say that the N Pacific may play a big role on long timescales. This statement seems to say the N Pacific also plays a role in G-IG timescales. I recommend omitting the "However" here.

Removed the "However".

Lines 197-199: I appreciate that the authors have toned down the carbon cycle part of this paper. Here, I happen to think this last statement doesn't have to be as guarded as it is. It would be fair to say that the changes in deep ocean circulation could potentially have significant impacts on the deep ocean carbon cycle.

Clearly this is the case, but what we were trying to communicate was that it doesn't necessarily follow that the same changes in future would have the same effect. Have changed the phrasing to reflect this.

Figure 3 Caption: "The Pliocene simulations are show relative to the pre-industrial experiment." Which direction was the difference taken? Plio-Pre-industrial or visa versa?

Added brackets to clarify this.

Figure 3: I appreciate that the authors have now indicated what the negative and positive values mean in MOC plots. But I would submit that if the authors want this figure to be meaningful to any non-modeler reading it they need to provide more help to the reader to understand how to interpret the anomaly panels. Seeing more contour lines on a plot generally indicates more not less. Explaining more clearly that in these plots more deeply contoured features (both positive and negative) mean a reduction not an increase in watermass strength because they are anomalies would be helpful.

Have added a sentence to the figure caption to lead the reader through interpreting the anomaly plots.

Figure 5: Include arrows that indicate ACC and Antarctic counter current?

Arrows added to Fig. 5 to depict these currents.